# Reception History and Early Chinese Classics

**Tobias Benedikt Zürn** 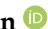

Department of Religion, Reed College, Portland, OR 97202, USA; tzuern@reed.edu

**Abstract:** Thus far, the study of early China and its texts is dominated by originalist approaches that try to excavate the authentic meaning of the classics. In this article, I promote the idea that a shift in focus from the intentions of the authors to the readers' concrete responses could meaningfully accompany our research on the classics' "original" meaning. Beyond merely illuminating the cultural and intellectual environments in which the various receptions were produced, such research on the classics' myriad interpretations could also serve as a postcolonial catalyst, helping us identify field-specific trends and reading strategies that, often unnoticed, impact our understandings of early Chinese texts. In other words, reception history would not only give us insights into the history of early Chinese classics and the variegated worlds they inhabited. It would also help us illuminate and reflect upon the ways we researchers shape and preconfigure our visions of premodern China and its texts.

**Keywords:** classics; reader response theory; reception history; *Laozi*; *Daode jing*; *Zhuangzi*; Daoism; originalism; philology; commentary

## 1. Introduction

The *Laozi* 老子 or *Classic of the Way and Virtue* (*Daode jing* 道德經) is one of very few premodern texts in world history that have garnered a truly global audience. As reflected in this Special Issue, it played a significant role throughout imperial China, peaking in the Tang 唐 (618–907) when Emperor Gaozu 高祖 (r. 618–626) claimed Laozi 老子, sometimes also called Lao Dan 老聃, to be the ancestor of the imperial Li 李 clan (Assandri 2022, p. 3). Moreover, the *Laozi* spread beyond the Middle Kingdoms (*zhongguo* 中國) due to its myriad translations resulting from an increasingly globalized world over the last few centuries (Tadd 2022a). In other words, it is a text with an impressive reception history. When one looks at the contributions in this Special Issue, one could thusly get the impression that Laozi studies or Laozegetics (*Lao xue* 老學) has been a thriving field in sinology. As Misha Tadd rightfully bemoaned, however, even though "scholarship on the *Laozi* is plentiful . . . mainstream research generally focuses on identifying the one 'correct' understanding of this work, with little recognition of its rich exegetical history" (Tadd 2022c, p. 1). Therefore, it is commendable to see this desideratum being addressed by a Special Issue, and its publication indubitably marks the beginning of a change in scholarly perspective on early Chinese classics like the *Laozi*.[1]

Because of this paradigmatic shift in studying early Chinese classics, I decided to write a short, programmatic piece engaging with the project's theoretical side rather than contributing another analysis of a concrete reception of the *Laozi*. In this essay, I will demonstrate how the study of early China and its texts could benefit from utilizing methods developed in the field of reception history, or reader response theory as it is more commonly called in the United States. I suggest that exploring the various interpretations of the Chinese classics as enshrined in commentaries, translations, and artistic re-inventions could favorably accompany our research on their "original" meaning beyond merely illuminating the cultural and intellectual environments in which the various receptions were produced (Tadd 2022c, pp. 10–11). In fact, it could also help us critically investigate how Eurocentric frameworks often operate unknowingly in the shadows of our argumentations.

I present this change of scholarly orientation in four steps. First, I provide a few examples of what I would term "originalist approaches" to the study of early Chinese classics; that is, scholarship whose focus lies in the excavation and retrieval of a text's "authentic" meaning.[2] In a second step, I suggest that this orientation assumes a clear boundary between a text and its readings. I propose, however, that such an approach that reads the various commentaries as strictly separate, since their annotations inevitably color our understandings of the classics, is neither historically evident through all stages of premodern Chinese history nor is it necessarily the only useful method at our disposal. In a third step, I therefore recommend adding reception history to our methodological apparatus. To account for this change in orientation, I broaden the definition of "text" as utilized by originalist scholars and introduce Karel Kosík's idea of "work." Kosík suggests that no cultural object or work is inherently infused with meaning that their creator(s) or author(s) left behind for us to excavate. Rather, meaning is generated by the continuous interaction between a work and its various audiences. Following Kosík, I propose that a reception historical approach to the classics would not only allow us to shine light on the various interpretive layers and biographies of the texts we study.[3] It would also enable us to gain a better understanding of our own positionality toward them. To substantiate this last claim, I paradigmatically showcase in the last part of the essay how reception history may provide valuable opportunities for self-reflection. By comparing the *Zhuangzi*'s 莊子 earliest reception with A. C. Graham's (1919–1991) evaluation from the 1980s, which still influences current engagements with the proto-Daoist classic, I demonstrate that at least some of the text's earliest readers—in my case Sima Qian 司馬遷 (ca. 145–86 BCE) and the authors of the *Grand Scribe's Records* (*Shi ji* 史記)—did not share Graham's evaluation of Master Zhuang or Zhuangzi 莊子 (fl. 4th century BCE) as a philosopher uninterested in politics and its mere quotidian concerns.[4] In other words, an evaluation of the *Zhuangzi*'s earliest explicit reception aside from the "All under Heaven" ("Tianxia" 天下) chapter reveals that Graham's reading of the proto-Daoist classic as first and foremost a philosophical text is less obvious than often assumed. Hence, the shift in focus from author-centered to reader-response-centered engagements with the classics bears the potential to offer eye-opening readings that may induce fruitful self-reflections on the history of Chinese studies and its institutionalized reading strategies. So let me begin this journey by providing a brief personal anecdote about how distinct disciplines perceive and engage texts differently.

## 2. How We Learn to Read Texts: A Personal View on Interpretive Communities

As an undergraduate student at Humboldt University in Berlin during the early 2000s, I was privileged to study in two distinct fields of area studies: premodern Scandinavian literature and sinology. I genuinely enjoyed the distinct knowledge I acquired during my study and my stays in Taiwan and Iceland. Particularly the experiences I gained during the two study abroad trips transformed my life and put me on the path of pursuing an academic career outside of Germany. In hindsight, however, I would say another element of my undergraduate education unsuspectedly had a major impact on my intellectual outlook. I realized firsthand during this formative period of my life how distinct academic fields train their students differently. At that time, I was not yet aware of Stanley Fish's work and his concept of interpretive communities that describes any scholar's inevitable embeddedness in a field of practice, so I did not have the terminological and intellectual apparatus at hand to grasp fully what I encountered (Fish 2001, pp. 36–38). Nonetheless, I was already quite aware that my two fields, Scandinavian studies and sinology, asked very different questions toward the texts we were reading and, in fact, had very different standards of what would comprise analytical evidence.

This distinction that I sensed in the early years of my college experience manifested most clearly in the training I received in both disciplines. Scandinavian studies, for example, introduced me to a postmodern canon of literary theory of which the majority was published in the post-1960s. This tendency was not surprising since the Institute of Northern European Studies (*Nordeuropainstitut*) at Humboldt University was shaped by scholars

like Bernd Henningsen, Stefanie von Schnurbein, Lann Hornscheidt, Kirsten Wechsel or Stephan Michael Schröder, who all focused on gender and cultural studies. I still remember vividly a conversation in Kirsten Wechsel's seminar that erupted after I presented an intimate love poem called "Hair" ("Hár") by Guðrún Eva Mínervudóttir, from the book *On the Brink of Pure Joy: Poems for Hrafn* (*Á brún alls fagnaðar: ljóð handa Hrafni*). The book contains two parts in which the two lovers Guðrún and Hrafn Jökulsson (1965–2022) wrote love poems to each other. In my presentation, I provided an author-focused reading of "Hár," arguing that it explicates Guðrún's attraction toward Hrafn's hair. My classmates attacked me for this interpretation, wondering why I was reading the poetic ego as the author and, more importantly, why I chose such an obvious, straightforward, and heteronormative piece for class discussion. In other words, I was critiqued for my approach to texts that emphasized an author's intent and a text's "original" meaning.

My experience in sinology was drastically different. The training in Chinese studies at Humboldt University's Department of Sinology, spearheaded by Florian Christian Reiter and Mathias Obert, focused almost exclusively on reconstructing the meaning of any given text at the time of its production. To achieve this goal, we received a very rigorous language training and were exposed to a few pragmatic aspects of philological analysis like the navigation of the imperially sponsored *Complete Library in Four Sections* (*Si ku quan shu* 四庫全書) or the consultation of commentaries as a means to decipher the meaning of a text. In other words, the intentions of the author(s) dominated and shaped our conversations about premodern Chinese writings, creating a stark contrast to the theoretical and methodological concerns I encountered in my Scandinavian studies courses. It instilled in us a vision of commentaries as separate and in service of the texts we read, sidelining, or perhaps even muting, any larger considerations of what we as scholars can do with writings beyond excavating their "original" meaning.

Of course, this experience was specific to my time as a student in Berlin in the early 2000s and should not easily and prematurely be projected onto any other place of higher education that offers these two fields of study. And more importantly, I do not suggest here a clear-cut hierarchy between theory and praxis as it is often displayed in the judgmental contrast between disciplines and area studies that one frequently encounters (Chen 2010; Davis 2015). In fact, I primarily consider myself a philologist that tries to combine elements of my text-critical and -analytical skillset with my training in comparative literature and cultural studies (Spivak 2003). Nonetheless, it was poignant that my experience in these two departments displayed such distinct approaches to texts, and I do think that my personal story at least partially illuminates a phenomenon that one repeatedly encounters in publications on premodern China: namely, the tendency in the study of Chinese classics to search almost exclusively for their "original" meaning and the intentions of their author(s), an ur-philological concern (Pollock 2014, 2015).[5]

## 3. Originalist Readings of the Chinese Classics

Let me substantiate this claim with the help of three cases. Take for example A. C. Graham's attempt at excavating the "original" teachings of Master Zhuang. Even though he admits that we scarcely know anything about this mysterious master beyond what is mentioned in the text named after him, Graham nonetheless tried to separate Zhuang Zhou's 莊周 authentic words from those portions that later authors presumably mixed into the text we read today (Graham 1981, pp. 29–30). In other words, Graham tried to identify Master Zhuang's voice in the extant *Zhuangzi* with the help of a careful philological analysis, so he may parse out the most "original" parts of the proto-Daoist classic from the rest (Graham 1981, p. 1; Liu 1994).

This type of philological dissection of classical texts into more or less authentic layers is typical for the sinological work that dominated the second half of the twentieth century. For example, E. Bruce Brooks and A. Taeko Brooks followed a similar path in their analysis of Kongzi's 孔子 (c. 551–479 BCE; latinized Confucius) *Analects* (*Lunyu* 論語). In fact, they aimed at reconstructing an "original" version of the Confucian classic in order to preserve

"an authentic glimpse of the historical Confucius" (Brooks and Brooks 1998, p. 1). According to this approach, philology would enable us to sift through the extant versions of the classic to identify and sort out the later insertions that hinder a clear understanding of the historical figure Kongzi and the intentions behind his teachings.

This heightened attention to authors' intended, "original" meanings that are hidden somewhere in the written traces of their teachings and therefore may be rescued from the classics' often messy textual formations and history (sometimes with the help of commentaries) may be traced back to both biblical studies and its impact on the humanities in Europe and the US, as well as the evidential scholarship movement (*kaozheng* 考證 or *kaoju xue* 考據學) from the Qing 清 dynasty (1644–1911). Both groups that seem to have impacted the onset of sinology and its philological orientation were particularly concerned with the reconstruction of authentic texts (i.e., the bible and the Confucian classics) in order to gain an unmediated access and unobstructed vision of Jesus and the sages.[6] Hence, it is not surprising that the study of early Chinese texts is dominated by an originalist approach that largely separates the various historical readings of these works from their "real" meaning.[7]

The focus on a text's "original" meaning rather than its historical interpretations was so dominant in the field of premodern Chinese studies that its framework even appeared in scholarship that generally would not share the same kind of originalist goals of dissecting the extant texts into more or less "authentic" remnants. For example, Stephen Bokenkamp's superb translation and discussion of the *Xiang'er* 想爾 commentary to the *Laozi* 老子 uses the same kind of argumentation to draw a clear distinction between the *Laozi*, which in Bokenkamp's assessment is not a Daoist text, and the *Xiang'er* commentary, which contains a Daoist interpretation of the proto-Daoist classic. As he remarks,

> The *Xiang'er* commentary is the earliest Daoist interpretation of the *Laozi* [and] the *Laozi* itself tells us nothing of the Daoist religion. Although the Celestial Masters accorded the *Laozi* primacy over other revealed texts as a catechism of their faith, their veneration seems to have been directed more to the figure of Laozi (or Lord Lao, as he was called) than to the ideas contemporaries found in the *Laozi* itself, for their interpretations often run counter to the clear intent of the text. (Bokenkamp 1997, pp. 29–20)

As we can see in this example, Bokenkamp uses the idea of a text's intent to distinguish between the early Chinese classic and the later interpretation of the *Laozi* by the Celestial Masters (*Tianshi dao* 天師道), the earliest Daoist community that settled in the region of modern-day Sichuan between the second and third century CE (Kleeman 2016). As Stephen Bokenkamp explained to me in an email conversation on 28 November 2022, he attempted "to translate the *Laozi* as the commentary suggested I read it. I thought that the Celestial Masters' readers would know the *Laozi* by heart, but my particular audience would not, so the separation of text from commentary was necessary." Accordingly, Bokenkamp materialized this concern in the page design of his translation, in which he visually disjoined the "classic" written in italics from the blocked off "commentary." Interestingly, this distinction of commentary and text is not present in the extant manuscript of the proto-Daoist classic housed in the British Library. There, the *Xiang'er* commentary presents itself as a text fully integrated in the *Laozi* (see Figure 1). Although it is possible that the early Celestial Masters community did not strictly separate between commentary and "original" text as suggested by the Dunhuang manuscript's textual design, it is also likely that differences in rhyme schemes, rhythm, content, and diction offered enough clues to distinguish between the two texts, particularly when read aloud.[8] Hence, it is understandable that Stephen Bokenkamp responded to this situation by comparing the manuscript with the various versions of the *Laozi* extant today to parse out the commentary from its main text. Even though the Celestial Masters might not have clearly separated the *Laozi* from their instructions and exegeses, Bokenkamp considered it nonetheless important to treat the "original" text and commentary as two distinct entities that need to be clearly distinguished.

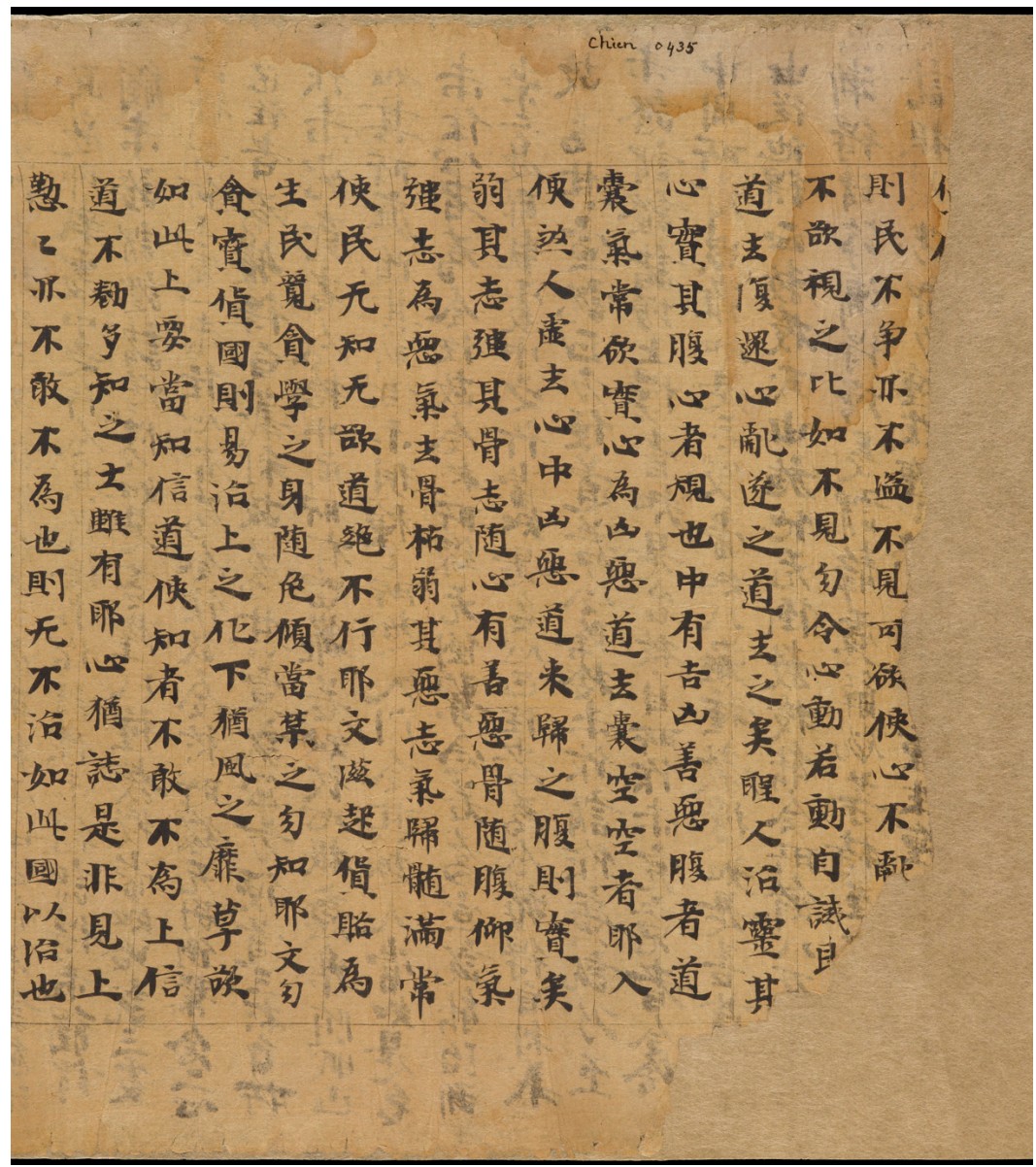

**Figure 1.** Beginning of the extant *Xiang'er Commentary to the Laozi* (*Laozi Xiang'er zhu* 老子想爾注) mainly displaying *Laozi* 3 and its commentary. Found in the Dunhuang caves and purchased by Sir Marc Aurel Stein (1862–1943). Photo courtesy of the British Library, Or.8210/S.6825, © The British Library Board. The title, main text and commentary are all written in the same style, blurring the boundary between main text and commentary.

Interestingly, his reason for distinguishing authentic and "original" layers of the *Laozi* from later additions is the exact opposite of Graham and Brook's goals. While the latter disentangle the early texts into layers to find the most valuable textual nuggets of an imagined authentic author, thus devaluing later layers as less significant contributions of later interpretive communities, Bokenkamp uses the distinction between commentary and main text to shine light onto the *Xiang'er* by de-emphasizing the importance of the *Laozi* and its intent for any understanding of early Daoist communities. Despite their differences though, the hard division between an "original" text and later additions—in the form of a commentary in Bokenkamp's case or "original" contributions by later communities "falsely" attributed to an early Chinese classic in Graham and Brooks' cases—seems inevitably to evince a more originalist mentality toward the classics that is not necessarily shared by premodern Chinese interpretive communities, to which I will return below.

All of the above-mentioned studies provided extremely valuable contributions to the fields of early and early medieval China. Particularly Stephen Bokenkamp's translation of the *Xiang'er* commentary into English offers an important alternative to Richard John Lynn's translation of Wang Bi's 王弼 (226–249 CE) commentary to the *Laozi* and Eduard Erkes' (1891–1958) dated rendering of the Heshang Gong 河上公 (fl. presumably 1st c. CE) commentary (Lynn 2004; Erkes 1945, 1946, 1949). However, we will see that originalist attitudes toward texts as demonstrated by Graham and Brooks should not be treated as the only meaningful engagement with the classics, since neither postmodern theorists, Renaissance humanists, nor premodern Chinese readers necessarily shared their orientation. In other words, it might be worthwhile to engage with Chinese classics beyond the scope of searching for a text's authentic and earliest meaning. To address this issue, let me briefly raise the question of what we consider a text in premodern China.

## 4. What Is a Text in Premodern China?

So far, I have suggested that the excavation of a text's essence, its authentic and "original" meaning, underlies a lot of scholarship on premodern China. Such originalist approaches, however, are far less natural and common throughout humanity's engagement with texts than many colleagues in the field of sinology assume. Postmodern theorists such as Roland Barthes (1915–1980) and Julia Kristeva, for example, aimed at decentering the author's claim to authority and the idea that a text's meaning must be rooted in an understanding that comes closest to readings prevalent at a work's time of production (Barthes 1977; Kristeva 1969). Particularly, their conceptualizations of intertextuality played a central role in weakening the importance of the author, since any given text is created in a web of cultural references that precede and at the same time exceed the personal and historical situatedness of its creator(s). As valuable as their contributions to the study of literature were, their concerns and ideas, however, developed during the cultural upheavals of the 60s and 70s. During this time, countercultural movements fought against traditional values including the romantic ideal of the author as a genius and *spiritus rector*, who embodies God's creative powers (Tomaševskij 2000). As a result, one might say that such postmodern visions would not fit a premodern context.

Interestingly, we find such anti-originalist approaches to texts not only in postmodern literary theory but also in premodern Europe. In the Renaissance, humanists who reconstructed ancient Greek and Roman classics often incorporated their commentarial and text critical work in the main text they were working on without marking these additions, blurring the boundaries between both texts and their interpretations, "original" authors and later editors. In fact, this practice was so common that writers such as Niccolò Perrotti (1429–1480)—like Graham and Brooks—publicly complained about the extant editions of classics, since contemporaneous humanists frequently inserted their own interpretations into manuscripts and changed the text (Grafton 2015, p. 165). Or as Anthony Grafton powerfully summarized, "the authors and the commentators were really all the same person" (Grafton 2015, p. 174). In other words, even though it might feel logical to clearly separate commentaries from their main text, I do think we should not assume that this separation is as universal and natural as it seems to many of us.

If we consider the context of premodern China, this separation becomes increasingly arbitrary. In fact, the cultural phenomenon of classics and their commentaries raises a sequence of questions that we oftentimes leave unanswered. What do we consider to be a text in premodern China? And was there a clear separation of text and commentary?[9] At first sight, these questions might seem nonsensical, since a commentary is an interpretation of a text and therefore should be treated as a separate, intentional reading. And in fact I do not try to simply conflate these two layers of a text. It is indeed often useful to read commentaries as separate from the main text. However, I would like to provide a few examples that pose some problems for our "obvious" distinction between main text and commentary.

First, premodern Chinese classics were almost always presented with and read through the lens of commentaries. Even though archeologists unearthed excavated manuscripts over the last few decades that do not contain commentaries, most classics like the *Spring and Autumn Annals* (*Chunqiu* 春秋), the *Book of Changes* (*Yi jing* 易經), the *Book of Songs* (*Shi jing* 詩經) or the *Laozi* were handed down and read in conjunction with one or multiple commentaries (*zhu* 注 or 註) and sub-commentaries (*shu* 疏) from the Han 漢 dynasty (206–220 BCE) onward. In fact, people throughout Chinese history rarely read the classics without the accompanied commentaries. Perhaps the most famous example of such exegesis based on commentarial traditions is the three schools of Kongzi's *Spring and Autumn Annals* (*Chunqiu san zhuan* 春秋三傳): the commentaries of Gongyang (*Gongyang zhuan* 公羊傳), Guliang (*Guliang zhuan* 穀梁傳), and Zuo (*Zuo zhuan* 左傳). As Anne Cheng claims, "The three extant commentaries must have stemmed originally from different schools of interpretation, and were the objects of passionate discussion during the Han dynasty, with each school of thought claiming to be the bearer of Confucius' authentic teaching" ([Cheng 1993](#), p. 68). According to Cheng, these schools did not only provide interpretations of the *Spring and Autumn Annals* but construed themselves as a powerful, perhaps even sole gateway to the "authentic" thoughts and actions of Confucius. Hence, commentaries were sometimes thought to be integral to "original" texts since only their mastery would provide exclusive access to the meanings and intentions of the main text's author(s). In other words, these traditions responded exactly in the opposite manner to Graham and Brooks. Rather than dissecting and liberating the main texts of anything "superfluous" to unearth and explore their "original" meaning, the three schools increased materials associated with the classics by creating and including later commentaries and additional passages.

Second, the difference between commentary and main text is sometimes hard to discern visually. Contrary to footnotes that place a comment in a space that is clearly marked as separate from the main text, writers in early China often embedded commentaries into the source texts they were discussing, as we have seen in Figure 1 above. Although they commonly interspersed the main texts with their summaries, glosses, and explanations, these posthumous paratexts, to use Gérard Genette's terminology, were not consistently differentiated by the size of their characters, as evinced in the contrast between Figures 1 and 2 ([Genette 1991](#), p. 264). This at times close relationship between commentary and main text might be the reason why we may find several passages in early Chinese writings nowadays that look like insertions of textual materials that previously might have belonged to commentaries—like the end of the *Zhuangzi*'s "Butterfly Dream" (*Zhuang Zhou meng hudie* 莊周夢蝴蝶 or simply *Mengdie* 夢蝶), which summarizes the short vignette in the style of an interlinear commentary by saying "This is called the Transformation of Things" (*ci zhi wei wuhua* 此之謂物化)" ([Guo 1954](#), pp. 53–54; [Mair 1994](#), p. 24).[10]

Third, the integration of commentaries into the main text was probably reflected in the conceptualization and understanding of texts (*wen* 文) as woven patterns (*wen*). For example, Liu Xie's 劉勰 (fl. late fifth century CE) *Patterned Hearts and Carved Out Dragons* (*Wenxin Diaolong* 文心雕龍), the first comprehensive and systematic treatise on early Chinese literary thought from imperial China, used weaving imagery to describe the "Ten Wings" ("Shi yi" 十翼) to the *Book of Changes*, a commentarial work traditionally attributed to Kongzi ([Liu 1978](#), p. 2). In this vision, the classic is a textual pattern that consists of Fu Xi's 伏羲 trigrams that function as the warp (*jing* 經) of the *Book of Changes* while the various comments attributed to Kongzi serve as the textual fabric's weft strands (*wei* 緯) that, read together, make the text's pattern crystallize. Liu Xie strongly contrasts his evaluation of the *Yi jing* and the "Ten Wings" with his assessment of Han Weft-Writings (*chen wei* 讖緯), which according to his estimation falsely claim that "they bear to the classics the same relationship that the woof bears to the warp in weaving" (蓋緯之成經，其猶織綜; [Liu 1978](#), p. 30; [Shih 2015](#), p. 27). In other words, the *Wenxin diaolong* seems in these two instances to respond to a common perception in early imperial China: namely, that texts, intertextual writing practices, and the production of commentaries were thought through and discussed in weaving terms ([Puett 2021](#), pp. 99–101; [Zürn 2020](#)).

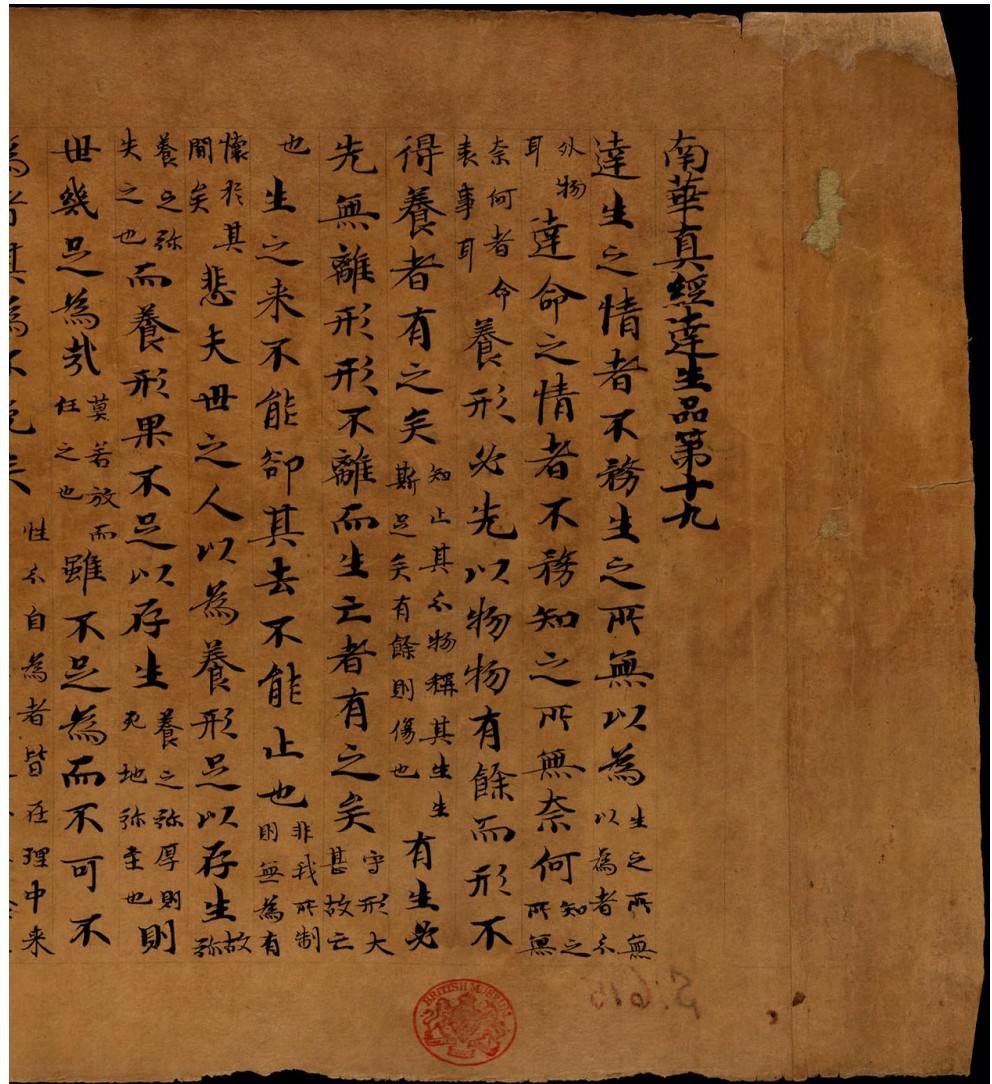

**Figure 2.** Beginning of the *Zhuangzi*'s chapter 19, titled "Understanding Life" ("Dasheng" 達生). Found in the Dunhuang caves and purchased by Sir Marc Aurel Stein (1862–1943). Photo courtesy of the British Library, Or.8210/S.615R, © The British Library Board. The title and main text are printed using large characters. Guo Xiang's 郭象 (c. 252–312 CE) commentary to the *Zhuangzi*, however, is printed in two columns using smaller fonts.

To summarize, none of these cases provide hard evidence that the separation between main texts and commentaries did not exist in premodern China. And I do not suggest that we should simply dispose of this separation. Of course, the blurring of the boundaries between commentary and main text might stem from practical considerations such as the necessity to save writing materials. Nonetheless, I think that the examples from early Chinese hermeneutic traditions, the material manifestations of commentaries, and contemporaneous conceptualizations of writings as textual fabrics as mentioned above should make us wonder whether the clear separation between main text and its annotations that dominates current engagements with early Chinese classics was evident throughout premodern China. Take, for example, Zhu Xi 朱熹 (1130–1200) who prominently bemoaned that his contemporaries were focusing so much on the commentaries and their exegeses of the sages' classical texts that they became "unacquainted with the master [i.e., the classics]" (Gardner 1990, p. 159). In fact, he complained that some of his contemporaries completely stopped reading the classics and based their understandings solely on commentaries. Hence, Zhu Xi's critique implies that there existed a debate on the value of com-

mentaries and their relationship with the classics during the Song 宋 dynasty (960–1279). And at least some readers, namely the ones who engendered Zhu Xi's strong criticism, apparently considered commentaries to provide exclusive access to the authors' intentions enshrined in the written traces of the classics.

The same attitude seems to be shared by the producers of a specific version of Cheng Xuanying's 成玄英 (fl. 7th c. CE) *Laozi* commentary. As Friederike Assandri remarks regarding the textual design of Dunhuang Manuscript P 2517 in her contribution to this Special Issue:

> Before the reader gets to see the first line of the base text, he has already read a structuring comment which relates the chapter to the previous one [and] an outline of the arguments the chapter will propose . . . In the Dunhuang manuscript P 2517 . . . these parts are in regular-sized characters, just like the cited base text. Only the interlinear commentary to the single lines is in smaller-sized characters. (Assandri 2022, p. 7)

Apparently, Cheng Xuanying's structuring comments were so integrated and strategically placed at prime positions of the main text that his voice could hardly be ignored. Hence, the examples of Zhu Xi's critique of contemporaneous reading practices, Dunhuang Manuscript P 2517, and the *Xiang'er* housed in the British Library show that commentary and classic were sometimes so interwoven that it almost seemed inconceivable for at least some premodern Chinese readers to point at the meaning of the classics without engaging in the interpretations offered by the commentarial traditions, a matter reflected in the fact that commentaries frequently refer to each other rather than the base text.[11] Hence, it seems as if we deal here with multiple, distinct attitudes toward the classics and their commentaries, and the originalist approach shared by Graham, Brooks, and apparently Zhu Xi is but one of many ways one may engage with these materials. Accordingly, I suggest it might be productive to think of a classic not just as a single (ur-)text whose one "original" meaning we must excavate. For the purpose of a reception history, it is even more fruitful to consider classics as multifarious cultural textures—what Kosík will call "works" in the next section—that accumulated various versions, diverse readings and reworkings over their existence.

## 5. A Shift in Focus: From Author- to Reader-Response-Centered Interpretations

If we accept for the time being that the distinction between author(s), main text, and commentary might not have always been clear and discernible (or we might say the distinction was less significant at times), it raises questions about how we might meaningfully engage with these texts. Clearly, we can read Graham and Brooks' work as an exclusionary or restrictive response to such a distinct understanding of authorship and text that might have been present in early China (Du 2018). The multi-layered textual formation of the *Zhuangzi* and *Analects*, as well as the existence of their many versions, apparently led these scholars to search for the one text that contains the "original" voices of the Warring States masters. In the remainder, however, I would like to suggest an alternative, more inclusive response that in my opinion can productively accompany the originalist approach: in some of our projects, we could de-emphasize authorial intent as the prime target of any hermeneutic enterprise and, instead, focus on a text's readership or what Sheldon Pollock termed "the second dimension of philology" (Pollock 2014, pp. 409–11). Such anti-originalist approaches were not only important for postmodern groups like Tel Quel, Renaissance humanists, and some premodern Chinese textual traditions, as suggested above. They were also formative for neo-Marxist understandings of cultural products. Karel Kosík, for example, emphasized the importance of the audience for the existence of any "work," that is, a cultural object created by human labor (Kosík 1976, pp. 66–77).[12] In his understanding,

> [The work] lives as long as its influence lasts. The influence of a work includes an event that affects both the consumer of the work and the work itself. What happens to the work is an expression of what the work is . . . The work is a work and

lives as a work because it *calls for* interpretations and because it has an *influence* of many meanings. (Kosík 1976, p. 80)

In this short passage from his early 1960s book *Dialectics of the Concrete*, Kosík describes the study of a work's "influence of many meanings" and reception history, a phenomenon that in English publications in the field of premodern China has received relatively little attention thus far.[13] Contrary to an institutionalized reading strategy in sinology that frequently judges divergent interpretations of a text as varyingly successful attempts at recapturing its "original" meaning, Kosík claims that differences in readings are concretizations of a work's internal powers that are reflected in readers' manifold responses to it.[14] Therefore, meaning is not only a crystallization of authors' intentions. One might rather say that readers repeatedly actualize a text by interacting and "working" with it. As Kosík claims, "[b]y outlasting the conditions and the situation of its genesis, a work proves its vitality . . . The work's life is not the result of its autonomous existence but of the *mutual interaction of the work and mankind*" (Kosík 1976, pp. 80–81). In other words, he perceives texts not to be static entities, that is, fossils whose "original" shape and meaning we try to excavate in the hermeneutic process. Rather, he treats them as analogous to living entities that realize themselves over time and therefore call for multifarious interpretations.[15]

Consequently, Kosík called for a paradigmatic shift in our understanding of the relationship between authors, their works, and their audiences. Instead of focusing on the ingenius intention behind a work, an idea that finds its roots in the Enlightenment movement and Romanticism's fetishization of the lives of authors, he emphasized the importance of the audience for the continuous recreation of a text's meanings (Tomaševskij 2000). According to his depiction, it is thus important not to reconstruct one authentic meaning but to explore people's concrete responses throughout various historical periods to any given work (Jauss 1982; Sarafinas 2022, pp. 8–11), a focus shared by several contributions to the Special Issue on Global Laozegetics (Assandri 2022; Constantini 2022; D'Ambrosio 2022; Gao 2022; Hadhri et al. 2022; Seo-Reich 2022; Tadd 2022b; Yao 2022; Zhang and Xie 2022; Zhang and Luo 2022; Zhu and Song 2022).

## 6. Conclusions: Why It Is Worthwhile to Explore the Reception History of Classics

One may wonder now why we should care what various audiences had to say about early Chinese texts. How does such an approach help us "better" understand the classics? Let me briefly present the potential value that the study of early Chinese texts through the lens of reception history may offer to us. To do so, I will paradigmatically discuss an example that is related to the international research project on the "Global Reception of the Classic *Zhuangzi*" (www.zhuangzi-reception.org) Mark Csikszentmihalyi and I founded in 2018. Despite excellent scholarly work on individual receptions, current engagements with the *Zhuangzi* in classrooms and journals around the world barely reflect the text's long-lasting influence (Hoffmann 2001).[16] Based on an assumption deeply rooted in Karl Jaspers' (1883–1969) vision of an Axial Age according to which the Han dynasty serves as a transition between the philosophical golden age of the Warring States period (475–221 BCE) and the rise and dominance of religious movements during the Six Dynasties (220–589 CE), most interpretations of the *Zhuangzi* in the last four decades followed A. C. Graham's influential assessment of the text that I introduced above (Graham 1981, 1989; Jaspers 1954; Roetz 1992). They focused on its "Inner Chapters" ("Neipian" 內篇), since these portions were thought to provide the most coherent and authentic picture of Master Zhuang's philosophy (Cook 2003; Kjellberg and Ivanhoe 1996; Mair 1983). In fact, Graham was so convinced Master Zhuang was a philosopher according to modern Western standards that he proposed, "the last of the *Inner chapters*, centered on a theme in which Chuang-tzǔ was hardly interested, the government of the empire," should be considered a flawed insertion of inauthentic materials since "one has an especially strong impression, not of an author approaching his topic from different directions, but of an editor going to great pains to find even remotely relevant passages" (Graham 1981, p. 29). As a result,

concerns that people nowadays would attribute to the fields of politics, arts, religion, and literature, to name only four other disciplines, have had little impact on Graham's reading of the *Zhuangzi*, which heavily impacted the classic's academic discourse over the last few decades, even though the long reception history of the proto-Daoist classic—and even the text itself—does not support such a dominance of one discipline over any other (Angles 2020; Hoffmann 2001; Meulenbeld 2012; Qiu 2005; Saso 1983). In other words, A. C. Graham provides here a case that substantiates Daniel Sarafinas' claim in this Special Issue: that "notions of authorship of a text influence, often unconsciously, a reader's interpretation such that the possible meaning generated within that text becomes limited, reduced, or terminated" (Sarafinas 2022, p. 1).

However, when we compare this trend with the reception of the *Zhuangzi* found in Sima Qian's *Grand Scribe's Records*, the earliest extant evaluation of the proto-Daoist classic, we see a quite different reading. In the *Shi ji's* "Biographies of Laozi and Han Fei" ("Laozi Han Fei liezhuan" 老子韓非列傳), Sima Qian claims:

> There was nothing on which his [i.e., Zhuangzi's] teachings did not touch, but in their essentials they went back to the words of Laozi. Thus his works, over 100,000 characters, all consisted of allegories. He wrote "Yufu" 漁父 (The Old Fisherman), "Dao Zhi" 盜跖 (The Bandit Zhi), and "Quque" 胠篋 (Ransacking Baggage) in which he mocked the likes of Confucius and made clear the policies of Laozi. (Sima 1994, pp. 23–24)[17]

> 其學無所不闚，然其要本歸於老子之言。其著書十餘萬言，大抵率寓言也。作漁父、盜跖、胠篋，以詆訿孔子之徒，以明老子之術。 (Sima 1962, p. 2143)

In this passage, the *Shi ji*'s reception of the *Zhuangzi* emphasizes three subsections all of which come from the "Outer" ("Waipian" 外篇) and the "Miscellaneous Chapters" ("Zapian" 雜篇). Thus, it seems as if at least the *Shi ji*'s authors were less concerned with the scant philosophical value of the supposedly less coherent portions of Master Zhuang's text as Graham propagated throughout his work (Graham 1981, pp. 27–39). Instead, some people in the Han apparently showed interest in political aspects and "inter-school" mockeries as the *Shi ji*'s account insinuates, attaching importance to different sections of the *Zhuangzi* beyond Graham and the modern assessment of the classic (Klein 2011).

In my opinion, this brief comparison concisely illustrates the importance of what Karel Kosík means when he says "what happens to the work is what the work is." The *Zhuangzi* seems to take on quite different lives in these two examples depending on which portions of the text an audience privileged—or even altered as in the case of A. C. Graham (Michael 2022, p. 4). On the one hand, the *Zhuangzi* and particularly its "Inner Chapters" appear to be, per Graham, a philosophical text engaging with questions of language and epistemology. On the other hand, Sima Qian presents the *Zhuangzi* and particularly its "Outer" and "Miscellaneous Chapters" as a text full of traces recording a polemical battle between two social groups associated with the figures Kongzi and Laozi. In other words, Graham's originalist readings of the *Zhuangzi*, and by extension the field's general orientation to texts, do not represent an objective approach to early Chinese classics. They are—like Sima Qian's interpretation—a historically contingent trend and a rather short episode of the work's long life story.

As a result, I am not calling for a postmodern revolution of the field or a complete dismissal of textual critical work and philological methods, since these approaches have yielded immensely valuable insights into premodern China and its texts. Rather, I suggest a rigorous diversification of our methodological apparatus (van Norden 2007, p. 6). In addition to the common practice of searching for the "original" meaning of a text, it would be beneficial to explore the various topics that guided historical interpretations of classics like the *Zhuangzi* or the *Laozi*. In so doing, we would not only learn more about the intellectual and cultural environments within which the classics' various interpretations were shaped, but we would also provide voice to all those traditional readings and practices that are

repeatedly marginalized in the academic discourse, since they do not conveniently fit into our visions of the classics (Denecke 2011).

Hence, reception history would allow us to reflect critically on our own thrownness, to use Hans Georg Gadamer's (1900–2002) terminology (Gadamer 1989). In contrast to its general reputation as a conservative method, it would grant us an opportunity for personal and institutional self-reflection that could unearth how our own, modern and oftentimes Eurocentric categories and divisions into academic disciplines secretly impact the way we read these early texts, as displayed in A. C. Graham's take on the *Zhuangzi*. Since each generation approaches the classics with its own concerns and frameworks, research into reception history is a prerequisite for a historically embedded understanding of these texts and our own interpretations. Thus, a focus on the classics' reception history would inevitably create more awareness regarding the fact that meaning is not simply inherent to a work or any of its envisioned authors. Rather it is generated in the power- and interplay between specific audiences, their cultural and historical horizons, and their texts. Or as Sheldon Pollock summarizes, "what a text means can never be anything but what the text has been taken to mean by the people who have read it. Its one true meaning can be nothing but the assemblage of all these other meanings . . . what the text may have meant to the first audience; what it meant to readers over time; what it means to me here and now" (Pollock 2014, p. 410).

**Funding:** This research received no funding.

**Institutional Review Board Statement:** Not applicable.

**Informed Consent Statement:** Not applicable.

**Data Availability Statement:** Not applicable.

**Conflicts of Interest:** The author declares no conflict of interest.

## Notes

1    In this article, I use the term classic in a wider sense than the Chinese term *jing* 經 is commonly used. In my understanding, it refers to any text that has accumulated a significant exegetical tradition in the form of commentaries, translations, and reworkings in various cultural products.

2    For a discussion of the same hermeneutic phenomenon, see the section "Authorial Intentionialism and Its Limits" in (Sarafinas 2022).

3    I use the term biography in relation to books since it reflects the idea that a text goes through different stages of existence, like human beings. The same vision is reflected in Princeton University Press' series "Lives of Great Religious Books." See https://press.princeton.edu/series/lives-of-great-religious-books, accessed on 10 December 2022.

4    I agree with most scholars of "religious" Daoism that we may only find a concrete community of people in the first and second century CE that formed a distinct group we may nowadays term Daoist. But unlike most scholars of early China or Michel Strickmann (1942–1994) and his students of later Daoist movements, who see a strict division between what scholars in early China oftentimes call early "philosophical" Daoism and later "religious" Daoism, I perceive a discontinuous continuity between these two "movements" in the form of shared terminologies, concepts, and practices. In other words, I follow Kristofer Schipper's (1934–2021) vision and call texts like the *Laozi*, *Zhuangzi*, or even by extension the *Huainanzi*, proto-Daoist, since they at least partially informed the lifeworlds and *imaginaires* of later Daoist practitioners.

5    Sheldon Pollock divides philology into three "dimensions": a text's genesis, its tradition of reception, and its presence to the philologist's own subjectivity (Pollock 2014). In my opinion, the first dimension outplays the other two in the field of early China.

6    For a discussion of "kaozheng-scholarship [as] a step toward indigenous development of an empirical mode of scholarship, even of modern science" (Quirin 1996, p. 36), see (Elman 1984). For a critique of readings that see the rise of modernity and scientific methods detached from ethical and moral concerns central for Confucian discourse in the Qing dynasty, see (Quirin 1996). For a discussion of the racist undertones of the purity discourse that guided the rise of the discipline of philology, see (Lin 2016).

7    Interestingly, rabbinic readings of the bible emphasize the multivalency of the text of which "multiple meanings [can] be derived from and are inherent in every [biblical] event, for every event is full of reverberations, references, and patterns of identity that can be infinitely extended" (Handelmann 1982, p. 37). I learned about Handelmann's work from (Wagner 2012, p. 65).

8    I would like to thank my colleague Alexei Ditter who reminded me that the performance and recitation of texts can enable an audience to experience stylistic differences between texts even if these distinctions are not reflected in the visual design of a manuscript. In that sense, separating commentary and main text on a visual level would be similar to the practice of adding

punctuation to early Chinese manuscripts: apparently, neither of these technolgies were needed by early audiences according to such a reading since they knew their texts by heart.

9  Hans van Ess argues that from the Han onward linguistic changes rendered the language of ancient classics so obscure to readers at the time that commentaries and phonetic glosses became a necessity for any engagements with the classics (van Ess 2009, pp. 216–25). Acording to Michael Puett, this attitude to commentaries as "the only source of access to the earlier material" changed only with Zhu Xi 朱熹 (1130–1200) in the twelfth century, whose orientation toward the classics I present on pages 8–9 (Puett 2021, pp. 105–6).

10  I would like to thank Mark Csikszentmihalyi who made me aware of this possible reading of the "Butterfly Dream's" coda.

11  As a graduate student at the University of Wisconsin, I met a colleague who displayed a similar take on the relationship between main text and commentary. In my first class on the *Zhuangzi*'s reception history in 2008, the said classmate repeatedly responded to the question of what is the meaning of a cryptic *Zhuangzi* passage by simply translating or summarizing Guo Xiang's commentary, effectively equating the main text with one of its interpretations.

12  For a radically different interpretation of the term "work" that reads it as the "receptacle of the Author's meaning" (Sarafinas 2022, p. 2), see (Barthes 1977).

13  There is a sizable amount of scholarship that could be categorized as studies in readings of early Chinese classics. However, very few of the examples mentioned in this footnote explicitly frame their work in such terms and engage with commentaries without referencing the field of reception history. For a few examples that engage with the reception history of the *Lunyu*, see (Ashmore 2010, pp. 111–97; Fuehrer 2002, 2009; Makeham 2003; Swartz 2008). For a few examples that engage with the reception history of the *Yi jing*, see (Schilling 1998; Smith et al. 1990; Smith 2008, 2012). For a few examples that engage with the reception history of the *Laozi*, see (Tadd 2022a; Chan 1991; Wagner 2000). For a few examples that engage with the *Zhuangzi*'s reception, see n.16 below.

14  For an example of a scholarly work that "shifts the emphasis from the author as the main creator and ultimate arbiter of a text's meaning to the editors and publishers, collectors and readers, producers and viewers, through whose hands a text, genre, or legend is reshaped, disseminated, and given new meanings" (pp. 1–2), see (Zeitlin et al. 2003).

15  For two projects that explore the varying images of Confucius, see (Csikszentmihalyi 2001; Nylan and Wilson 2010).

16  For a few examples of excellent work on the *Zhuangzi*'s reception, see (Angles 2020; Brackenridge 2010; Chai 2008; Chapman 2010; Choi 2010; Epstein 2006; Fang 2008; Harack 2007; Idema 2014; Liu 2016; Möller 1999; Qiu 2005; Saso 1983; Saussy 2017; Specht 1998; Swartz 2018; Tang 1983; Wang 2003; Xiong et al. 2003; Yu 2000; Zhang 2018; Ziporyn 2003).

17  I changed the transliterations from Wade-Giles to Pinyin in this quotation.

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
