# Peer review of "Reception History and Early Chinese Classics"

_religions, doi:10.3390/rel13121224_

Round 1

Reviewer 1 Report

The author presents a convincing argument for the merit of commentarial literature and its research-worthiness. Indeed, this paper, once published, will help promote such research and may create a new turn in Western sinological studies on Chinese philosophy. 

On the whole, I agree with virtually everything said in the paper, but one point might be worth reconsidering. The apparent lack of differentiation between main text and commentary (ll.279-286) may seem so only to modern readers as it is very likely that the users of the text were aware of the distinction. Moreover, the main text and commentary were blended together perhaps only for the sake of convenience and necessity (e.g., to save paper). 

Finally, the author's language and style are perfectly fine but there is no such option on the question.

Author Response

Dear Reviewer 1,

Thank you very much for your kind review. I am very glad you liked the article. Here come my responses to your two points of critique.

1) I added note 8 in response to your claim that "The apparent lack of differentiation between main text and commentary (ll.279-286) may seem so only to modern readers as it is very likely that the users of the text were aware of the distinction." I mention in the footnote "that the performance and recitation of texts can enable an audience to experience stylistic differences between texts even if these distinctions are not reflected in the visual design of a manuscript." I also toned down my claim throughout the essay (see lines 335-345 in particular).

2) Regarding the claim that "the main text and commentary were blended together perhaps only for the sake of convenience and necessity (e.g., to save paper)": I added a sentence in lines 337-339 that addresses this claim. In my opinion, technical limitations do not fully explain though why Cheng Xuanying's commentary, as discussed in Friederike Assandri's article, would be integrated in the main text. If the writers of Dunhuang manuscript P 2517 (and the Xiang'er) would have wanted to save paper, they would have written Cheng Xuanying's summarizing comments in small fonts.

Thank you again for your help!

Reviewer 2 Report

In this erudite yet synoptic paper, the Author challenges the dominant strategies of interpreting premodern Chinese philosophical texts. By using Laozi as a case study,  she/he convincingly contests the stereotypical division into 'original' treatise and its 'secondary' commentary and demonstrates the methodological pitfalls entailed by the focus on (or rather quest for) mythical authorial intent. Most importantly, the paper does not suggest that the proposed framework should replace current approaches, but rather enrich them and add something to them in the spirit of methodological pluralism, especially when their hermeneutic efficiency is way below our noble expectations. Therefore, the paper is a significant contribution not only to "Laozegetics," but also to the general methodology of Chinese philosophy.

Author Response

Dear Reviewer 2,

Thank you very much for your kind review. I am very glad you liked the article. It is very encouraging that you fully endorsed the project, and I very much appreciated your supportive comments.

Wish you all the best.

Reviewer 3 Report

This is a very interesting Essay that questions the way early Chinese texts are interpreted in the West. In particular, the Author makes a very good point in relation the A. C. Graham’s division of the Zhuangzi. Although I am very sympathetic with the Author approach, and I see some merit in their focus on the “Reader-Response-Centered Interpretations” I have some concerns in relation to the philosophical argument that the essay presents.

Besides the fact that the main example is the Zhuangzi and not the Laozi – which makes me wonder why this contribution should be include in this special issue – the real problem of the essay is that it equates the problem of the original text with the problem of authorship. This may be true for Graham’s approach to the Zhuangzi – and in this I agree with the Author that we need to avoid the distinction of the text in inner, outer, and miscellaneous chapters – but this does not mean that different versions of a text do not refer to different periods of time and different contexts ­– which should be analysed on their own before considering them as one hypertext.

The most philosophically problematic aspect of the essay is when it discusses the issue of the commentaries. One of the main points the Author makes in relation to this is that “the difference between commentary and main text is sometimes hard to discern visually.” (7) This is a very weak argument which seems to suggest that because the difference between the two texts is hard to discern visually, then we should not discern them, which implies that we should not make any distinctions between, for instance, the Guodian Laozi and its commentaries.

The fact that the Author has a confused idea of the issue is revealed in the last quotation where they write that “as Sheldon Pollock summarizes, “what a text means can never be anything but what the text has been taken to mean by the people who have read it. Its one true meaning can be nothing but the assemblage of all these other meanings ... what the text may have meant to the first audience; what it meant to readers over time; what it means to me here and now” (Pollock 478 2014, 410).” (12-13) This is certainly true, but it does not mean that the text does not have different meanings in different times and that different commentaries have different approaches to a common text of reference. Thus, even if we avoid the idea of an original text ­– and I agree that we should avoid it – it does not mean that there is no distinction between a reference text and a commentary or between different versions of the same reference text.

Having said that, I think that this can be a valuable article, provided that the Author gives a more robust theoretical account of their theory. As it stands, the essay suggests a very vague and contradictory idea of textual analysis.

On a minor note, in the Bibliography David Chai should be Chai, David.

Author Response

Dear Reviewer 3,

Thank you very much for your review and your general support of the project. I am glad you found my essay interesting and saw some merit in the reception historical approach. In the following lines, I will only respond to your points of critique, but let me emphasize at the beginning that your comments helped me tremendously in revising my paper. Thanks! Here come my responses:

1) Regarding the Zhuangzi and Laozi issue: I understand the concern, but I disagree that the paper does not fit to the special issue, just because its final example is taken from the Zhuangzi. The essay generates a justification for the type of work the special issue offers, extending the scope of this project from one proto-Daoist text (the Laozi) to our general approaches to classics. By providing a concrete example from the Zhuangzi's reception history (in addition to several points related to the Laozi), the article makes the claim that the type of work displayed in this special issue is relevant for scholars beyond the focused theme of Global Laozegetics. As I say at the beginning, the essay wants to point out that the special issue "marks the beginning of a change in scholarly perspective on early Chinese classics" (lines 36-37) and not only the Laozi.

2) The problem with connecting ideas of originality with the problem of authorship: I do not claim that these two problems are inherently connected, but that the "originalist approach" of scholars like Graham and Brooks equated "originality" with an imagined "author." In other words, the combination of "author" and "original" meaning is a feature of originalist approaches and not my understanding of "literature" or how we "should" read texts. My own understanding is perhaps best described by the quotation from Pollock at the end of the essay and by the following sentence: "it might be productive to think of classics not just as a single (ur-)text whose one “original” meaning we must excavate, but also as a multifarious cultural texture—what Kosík will call a “work” in the next section—that consists of the accumulation of all of its versions, diverse readings, and reworkings" (lines 364-367). In my opinion, the latter reading seems to be in line with reviewer 3's position who is interested in the historical contingency of the classics' various interpretations.

3) Regarding the idea that different versions of a text refer to different periods of time and different contexts: I think reviewer 3 misunderstood the project in this case.  For a more detailed explanation and a brief outline of my revisions, see point 5 below.

4) Regarding my discussion of commentaries: I disagree that the visual component of commentaries is my main argument. I mention three observations about commentaries, all equally important (although I would say the first observation is the most important one if somebody would force me to make a choice), to simply raise a question: is the separation between main text and commentary as obvious as we commonly assume. I mention throughout the article that I "do not try to simply conflate these two layers of a text" (line 271); I simply ask what if this separation was not always as important to an audience as it is for us now (see lines 369-373)? And would there be any consequences for our engagements with the classics if premodern audiences did not always share the same originalist approach (that is, the idea that a text's original meaning can only be found in the main text by peeling off any later interpretations) and vision of "text" that defines scholarship on early China in the 20th and 21st centuries? What if some people thought that the "original" meaning of a classic was only accessible via commentaries? Should we then still think of the commentary as a separate text? Or does it become integral and almost "part" of a main text if it was thought to be the sole gateway to a text's meaning? I mainly use this part of the article to question some of the assumptions that underlie originalist approaches to the classics. It simply prepares the ground for a change in scholarly orientation that emphasizes the importance of the various exegetical traditions for a historically embedded understanding of the classics. In any case, I have rewritten parts of my argumentation to improve the clarity of my writing and softened my tone throughout section 4 (particularly lines 339-350). Moreover, I included some explicit comments such as "I do not try to simply conflate these two layers of a text. It is indeed often useful to read commentaries as separate from the main text" (lines 271-272) so that my essay hopefully won't produce the impression anymore that I simply conflate commentary and main text.

5) Regarding the claim that I am confused and do not realize that texts
"have different meanings in different times and that different commentaries have different approaches to a common text of reference": I think that reviewer 3 was misled by my writing since I make exactly the claim that classics have multifarious meanings in various times throughout the entire article. In fact, my article consistently claims we should not only try to find the "one," "true" meaning of a text but engage with its various, historically contingent interpretations. Here come a few lines from the article that support my claim: "reception history would not only give us insights into the history of early Chinese classics and the variegated worlds they inhabited" (lines 13-15); "exploring the various the various interpretations of the Chinese classics as enshrined in commentaries, translations, and artistic re-inventions could favorably accompany our research on their “original” meaning beyond merely illuminating the cultural and intellectual environments in which the various receptions were produced" (lines 44-47); "a reader-response-centered perspective that enables us to explore the various layers of the classics’ biographies" (lines 58-59); "it is thus important not to reconstruct one authentic meaning but to explore people’s concrete responses throughout various historical periods to any given work" (lines 410-412); the entirety of section 6. To avoid misunderstandings, I rewrote parts of section four included phrases that explicitly mark my own position and agenda (see lines 335-348).

6) I fixed the entry in the Bibliography on David Chai.

Thank you for your careful reading and helpful comments. I hope I addressed most of the claims appropriately and look forward to hearing from you.

Wish you all the best.

Round 2

Reviewer 3 Report

I appreciate the effort the Author put into the revised version of the essay. In particular, I appreciate the detailed cover letter in which they address my comments. I find that some clarifications are useful to clarify the Author’s Position. Still, I struggle to understand the Author’s theoretical approach. If the point is to “recommend that now and then we might want to shift focus to a reader-response-centered perspective that enables us to explore the various layers of the classics’ biographies”, I do not see how this “now and then” can be implemented in a sound methodological procedure; that is to say, how do we decide to use one approach instead of the other? This is not clear. I would encourage the author to define this passage more attentively – this can help the overall quality of the essay's theoretical stance.

Author Response

Dear Reviewer # 3,

Thank you very much for your comments and continuing support. I have rewritten parts of the introduction and the passages in the sections that address my understanding of texts. I hope that these changes made clearer my agenda and theoretical framework.

Beyond these changes, I would like to address your concerns about "how do we decide to use one approach instead of the other: In my opinion, different methods allow us to explore different kinds of aspects of the classics. In that sense, there is no imperative to decide between any of them, may they be an originalist or a reception historical approach. It is simply a matter of one's research goals. I just suggest in my essay that reception history is a really good method for reflecting critically on the "reading strategies" of one's field.

Hence, my essay addresses a question of methodological diversity, which we scholars of early China do not really have in comparison to other disciplines. For example, I have seen various colleagues getting shut down by people who were insisting that it is only important to explore what any given text meant at the time of their production. People just rejected these presentations since they were asking different questions and were discussing different sources. My essay does not want to tell people who are concerned about originalist approaches that their work is unimportant (at least I hope that my essay does not create this impression), but that a reception-focused approach might provide interesting insights in how we scholars are entangled into our historically contingent visions of early Chinese texts that might be servicable to their goals.

Accordingly, my essay does not develop a "sound methodological procedure" but is an invitation to colleagues who are intrigued by the questions I raise to join our endeavors and explore the field of reception history. In the style of Hans Blumenberg who developed almost all of his theoretical frameworks in a paradigmatic way, I work out one example of a reception historical project and its value, in this case my critique of Graham and his reading of the Zhuangzi vis-a-vis the Shiji's interpretation, and gesture toward standard works in the field of reception history for all those who would like to become more acquainted with this methodology. I understand your concerns, though, and I will address them in a separate article on reception history as a critical method that I am writing for Critical Inquiry. I just think that answering your concerns about a theoretical framework are beyond the scope of this short essay. If you're interested, though, I will gladly share the results with you when I will probably finish this piece in Winter/Spring 2023/24.

Thank you again for your great help!